# ENTROPY-ENHANCED CONTEXT-AWARE EVENT PREDICTION BASED ON ONTOLOGY AND EXTERNAL KNOWLEDGE

## ABSTRACT

Predicting impending events is an attractive task of natural language processing (NLP), which plays an important role in many fields such as emergency management. Current predominant event prediction method is to construct event graphs and learn event representations through graph neural networks (GNNs), which mainly utilizes the semantic and structural information of events to obtain subsequent events, but ignoring the context of event. Meanwhile, these methods does not address the issue of heterogeneity of nodes and edges in event networks. Last, the lack of high-quality event datasets is also a challenge for event prediction. In response to the above issues, this paper proposes the **E**ntropy-enhanced **C**ontext-aware **O**ntology-based model (**ECO**), which introduces the Entropy calculation module to learn the heterogeneity of nodes and edges, thereby better learning event representations. Furthermore, external knowledge is introduced to the event graph to enhance the semantic information of events during the prediction. Finally, we design a context-aware event ontology for maritime emergency management, and construct a real-world dataset, **M**aritime **E**mergency **E**vents **D**ataset (**MEED**), to verify our prediction method. Experiments on node classification and link prediction show effectiveness and practicability of our proposed model in realistic scenarios.

## 1 INTRODUCTION

Events are something that happens with a specific participant, action, object, time and location. Understanding relationships behind events and making precise predication is one of the most important research branches of NLP.(Du et al., 2022)

Initially, event prediction is based on event pair (Chambers & Jurafsky, 2008) and event chain (Lv et al., 2019). Event-pair-based methods use the statistical information of event pairs to measure the relationship between events, while event-chain-based methods focus on the order of events. Since Glavas & najder (2014) proposed event graph, which contains events, event attributes and correlations between events, GNNs has been widely applied into event prediction.

However, most works are conducted on homogeneous graphs, ignoring the heterogeneity of event graphs. Unlike homogeneous graphs, which are graphs with only one type of node and one type of relationship (see Figure 1.a), the sum of the types of nodes and edges is greater than two in heterogeneous graphs (see Figure 1.b). In addition, existing studies on heterogeneous graph only focus on the heterogeneity of nodes or edges, not considering the case that both of them are heterogeneous. Furthermore, current approaches merely rely on basic event elements, such as people and objects involved, which ignore a lot of relevant details

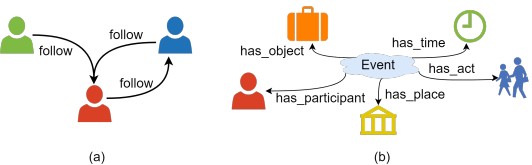

Figure 1: An illustration of homogeneous graphs and heterogeneous graphs. In (a), there is only one node type and one edge type. (b) is an event-centric event graph with six different kinds of nodes and five different kinds of edges.

that are conducive to event prediction. Take an car accident for instance, weather, road condition and other factors are all instructive for predicting the probability whether an event might occur.

To address these challenges, we proposed **E**ntropy-enhanced **C**ontext-aware **O**ntology-based model (**ECO**), which use R-GCN as backbone to predict maritime emergency event. To tackle dual heterogeneity of nodes and edges, we introduce entropy calculation module to learn the representation of events. We additionally used OpenConcepts as external knowledge to enrich semantics and build high level relationships between nodes. To test the validity of our method, we collect real-word maritime emergency reports and establish a **M**aritime **E**mergency **E**vents **D**ataset (**MEED**), which is build on the ground of a context-aware maritime event ontology. Finally, we verify **ECO** on **MEED**, and experimental results show that our model outperforms the original one, and ablation studies indicate that applying context-awareness, entropy, event ontology and external knowledge can yield better performance on representation learning of event prediction.

Overall, the main contributions can be summarized as follows: First, we propose a novel model, **ECO**, which introduce entropy calculation module to deal with the dual heterogeneity in event graphs and learn better event representation; Second, we devise a context-aware event ontology, which integrates realistic context into event graph, to depict the event comprehensively and to optimize event representation; Third, we establish a maritime dataset, **MEED**, to validate the performance of our proposed method in event prediction; Fourth, we conducted a series of experiments and ablation studies to confirm the effectiveness and practicability of our work.

## 2 RELATED WORK

**Event prediction with Graph Neural Networks.** Event graph, which is inspired by knowledge graph, can depict event elements and relationships between numerous events at the same time. (Li et al., 2018b) Li et al. (2018b) constructed a Narrative Event Evolutionary Graph (NEEG) based on the event chains, and proposed a Scaled Graph Neural Network (SGNN) to model event interactions and learn event representations. Huai et al. (2023) constructed a spatial-temporal knowledge graph and proposed a continuous-time dynamic graph neural network to predict events. Liu et al. (2022) proposed a End-to-End joint model for Event Factuality Prediction, using directional and labeled syntactic information graph to enhance graph neural network. Luo et al. (2020) proposed to use dynamically constructed heterogeneous graph to encode events. Wang et al. (2023) used R-GCN (Schlichtkrull et al., 2017) to predict associations of microbes with diseases. Although the existing works have shown effectiveness, they overlooked the fact that both nodes and edges in the event graph are highly heterogeneous. Therefore, our work focuses on the representation of dual heterogeneity of event graphs.

**Event prediction with ontology.** Ontology origins from philosophy, and it is defined as a systematic description of objective things in the world. (Hao et al., 2023) At present, many studies have introduced event ontology into event prediction, such as using event ontology to predict the event classification of basketball games (Wu et al., 2020) and sound (Jiménez et al., 2018). Zhuang et al. (2023) used event ontology to enhance the extraction of event relation, and to predict missing event information. Mao et al. (2021) constructed an emergency event ontology to characterize the objective laws between event evolution, and based on the ontology, they built a pipeline model to extract events and predict events. Lv et al. (2021) proposed a Web service recommendation method based on event ontology. By referring to hierarchical information in the event ontology, the semantic relationship between the event classes was calculated to predict users' need. Neumann et al. (2022) proposed a surgical process ontology and presented a prediction method to understand the situation in the operating room at any time and predict the following possible events. Existing event ontologies only cover the basic information of events, such as participants, time and place, without paying attention to the context when an event occurs. Therefore, we propose a context-aware event ontology to help to learn event representation.

**Event prediction with external knowledge.** Since knowledge can enhance interpretability for models, adding external knowledge in NLP tasks has been prevailing these days. For example, Lee et al. (2009) proposed a knowledge-based prediction model to discover traffic patterns and transformed them into rules for traffic event prediction. External knowledge can also be used to predict health events, such as the relapse of acute lymphoblastic leukemia patients. (Porzelius et al., 2011) At present, a majority of approaches are retrieving related entities directly from the corresponding

knowledge base. Huang et al. (2018) retrieved external knowledge and integrated it with Recurrent Neural Networks (RNNs) and key-value memory networks to predict users' preferences. Lv et al. (2020) used event elements, such as activities, states, events and relations to retrieve related knowledge, and then assembled them into the inference model to help predict the next event. Swati et al. (2022) adopted the method of translation and retrieval to obtain commonsense knowledge, then making prediction based on it. Deng et al. (2019) retrieved external knowledge from knowledge graph to obtain event embeddings and then combined event embeddings and price values together to predict stock events. We also adopt a retrieval approach, but we do not use external knowledge to interpret entities, but to establish relationships between entities, so as to make the representation of similar events more semantically interrelated.

## 3  METHOD

### 3.1  NOTATIONS

#### 3.1.1  EVENT

In order to describe events in a more distinct and hierarchical form, we define events from two perspectives, intra-event relationship and inter-event relationship.

**Intra-event relationship.**  An event is an act happens at certain time and in certain place, with specific agents and patients. Thus, an event can be defined as a quintuple formally (Liu et al., 2016):

$$Event = (Participant, Act, Object, Time, Place) \tag{1}$$

Participants are those who take part in the event; an act is the core of an event and is served as the trigger in event extraction task; objects are the objective things involved in the event; time can be a moment of a period of time when an event took place; Place is the location of an event happens. These elements in the five-tuple depict the relationship within a single event, such as $(eventA, has\_Act, sink)$ or $(eventB, has\_Time, 2020/8/23\ 09:50)$. These event elements, are viewed as the foundation in a majority of event prediction tasks.

**Inter-event relationship.**  Further, we define three relationships to describe the interaction between events. There are listed and defined as following:

(1) FOLLOW relation: is_follow means events are in time order. If event happens, and after a period of time, event happens, then there is a is_follow relationship between $eventA$ and $eventB$;

(2) CONCURRENT relation: If event and event happen simultaneously, then there is a concurrent_with relationship between $eventA$ and $eventB$ ;

(3) CAUSE relation: If the occurrence of event will directly lead to the occurrence of event , then there is a cause relationship between $eventA$ and $eventB$. Noted that although CAUSE relation is also chronological, but it emphasize causality between the two events, which is not present in is_follow and concurrent_with.

#### 3.1.2  CONTEXT

When humans predict an event, they are prone to pay more attention to the context of event rather than the status of the event (event elements). Thus, it is necessary to introduce context awareness into event prediction. Context awareness is widely used in recommendation system (Ashley-Dejo et al., 2015), sentiment analysis (Irfan et al., 2019), Internet of Things(IoT) (Arrotta et al., 2023), and so on. We notice that the importance of context in event prediction is neglected, so we view context as a part of the event, and introduce context elements into our maritime emergency event ontology. Context is generally defined as the situation within which something exists or happens and its definition may differ according to practical field.(Sezer et al., 2018) Schilit & Theimer (1994) described context as locations, identities of nearby people, objects, and changes to those objects, while Abowd et al. (1999) give context a more general definition: a context is the information to characterize the situation of an entity.

Here, we follow Chihani et al. (2011) and Nicolas et al. (2011), sorting event context into three types: entity information, social environment and physical environment. Like the event element, we

can denote it as a threefold-tuple:

$$Context = (Entity\_Information, Social\_Environment, Physical\_Environment) \quad (2)$$

### 3.1.3 CONTEXT-AWARE EVENT ONTOLOGY FOR MARITIME EMERGENCY

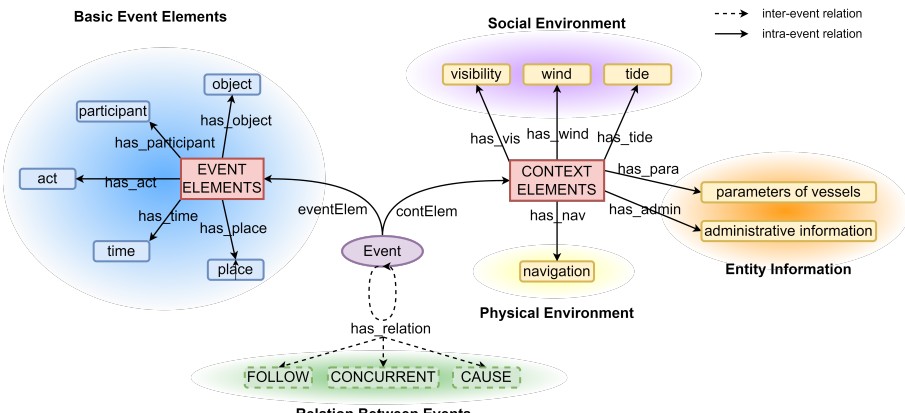

Figure 2: An illustration of event-centric event ontology for context-aware event graph.

Since the definition of context is a general one, it should be specified for different event types. In terms of maritime emergency prediction, entity information are attributes associated with the entities, such as parameters of vessels or administrative information about venues. Similarly, based on nautical expertise, social environment for maritime emergency prediction refers to the navigation condition, while physical environment refers to meteorological and hydrological characteristics. For instance, when a ship is wrecked on the rock, under context of a gale and a rainstorm, the probable subsequent events are more likely to be sinking or personal injury, while under context of an agreeable climate, the probable subsequent events can just be loss of economy. It is clear that difference between wind strength can lead to different consequence of varying severity, so it is crucial for event prediction.

At present, there are various methods to model contexts, such as key-value models, logic based models and graphical models. The structure of information is obscure, if the model simply relying on event triples and context triples, which is not conducive to learn the representation of different events. Thus, we do not take the above approaches, but devise context into event ontology. Event ontology is a shared, formal and explicit specification of an event class system model that exists objectively. With event ontology, semantic can be easier to share between entities, and it is also beneficial for strengthening representation and facilitating knowledge reuse. The hierarchy of our event-centered context-aware ontology for maritime emergency events is illustrated in Figure 2.

## 3.2 ENTROPY-ENHANCED CONTEXT-AWARE ONTOLOGY-BASED MODEL (**ECO**)

The architecture of **ECO** can be divided into five parts (see Figure 3).

**Context-awareness Fusion**. According to the definition of context we mentioned in 3.1.2, we extract context for every event and record them as triples. Then, we fuse them with the event graph.

**External Knowledge Integration.** After, social environment and physical environment are collected in Context-awareness Fusion module, entity information is acquired through external knowledge database. Here, event elements as used as queries to be retrieved in the knowledge base, and the results returned are the triples relevant to the query. With triples, we can build a knowledge graph (KG) in the same way we build the event graph, then the KG is integrated into the event graph as well.

**Entropy Calculation.** There are various way to calculate graph entropy, such as structure entropy, which is proposed by Li & Pan (2016) and has been used in network security(Li et al., 2016), genome informatics(Li et al., 2018a) and other fields. Scholars such as Rashevsky (1955), Trucco (1956), Mowshowitz (1968) and Körner (1973) also their own definitions of graph entropy. We

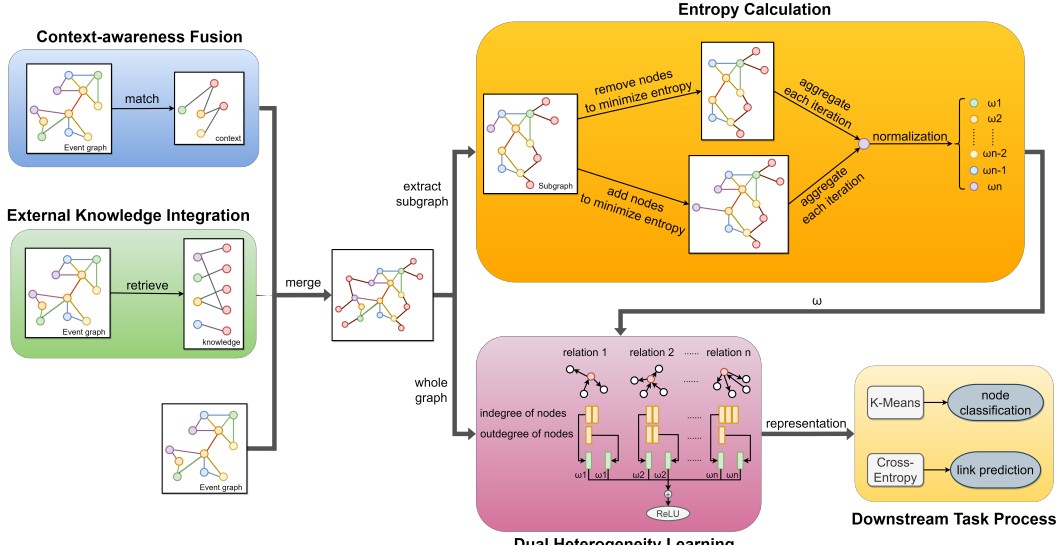

Figure 3: The architecture of **ECO**. The input of **ECO** is an event-centric event graph. Firstly, each event was associated with its corresponding context elements. Secondly, the event elements were retrieved in the knowledge base, and triples returned are used to form a knowledge graph. Thirdly, the heterogeneity of edges was learned through the entropy calculation module to obtain the weights. Fourthly, these three graphs are integrated into one and go through the dual heterogeneity learning module to get representations, and the weights obtained in the previous step are utilized in aggregation stage. Finally, execute node classification and link prediction tasks respectively.

follow Kenley's work(Kenley & Cho, 2011), defining graph entropy based on Shannon entropy. To resolve the quantitative measurement of information, Shannon first give out the definition of entropy as:

$$H(p_1, ..., p_n) = -\sum_{i=1}^{n} p_i log_2 p_i \tag{3}$$

Suppose the graph is undirected and without weight, we denote it as $G(V, E)$. $G$ can be decomposed into a set of clusters. A cluster is considered an induced subgraph $G'(V', E')$ of $G$. $G$ has dense connections within $G'$, while there is sparse connection between $G'$ and $G - G'$. Now given a subgraph $G'$, we define the inner links of a node $v$ as the edges from $v$ to the nodes in $V'$. The outer links of $v$ are defined as the edges from $v$ to the nodes not in $V'$. $p_i(v)$ denotes the probability of having inner links:

$$p_i(v) = \frac{a}{|N(v)|} \tag{4}$$

where $|N(v)|$ is the total number of neighboring nodes of $v$, and $a$ is the number of the neighboring nodes of $v$ that are in $V'$. Similarly, $p_o(v)$ denotes the probability of $v$ having outer links:

$$p_o(v) = 1 - p_i(v) \tag{5}$$

Node Entropy is defined as the probability distribution of inner links and outer links of node $v$, and we denoted as $e(v)$:

$$e(v) = -p_i(v) \log_2 p_i(v) - p_o(v) \log_2 p_o(v) \tag{6}$$

Then, we can defined graph entropy as the sum of the entropy of all nodes in $G$:

$$e(G) = \sum_{v \in V} e(v) \tag{7}$$

The pivot to calculate the entropy is to minimize the graph entropy by adding or removing nodes in a subgraph, the algorithm of which is concretely presented in Appendix A.

Subsequently, we collect the node entropy calculated in each iteration. If two nodes are connected by an edge, then we sum the entropy of these two nodes to represent the edge entropy. For example,

if there are relationship (A, is_a, B) and (C, is_a, D), the sum of the entropy of A, B, C and D is the edge entropy of is_a. Finally, we normalize the entropy of different type of edges as weights, which characterizes the dual heterogeneity of nodes and edges, so as to help learn a better event representation in the next module.

Essentially, what this module learns is the inner structure of the event graph, which can be considered as a kind of internal knowledge. In addition, since the entropy is calculated according to a specific event graph, the result may differ due to the changes in event graph, which means that our model can deal with dynamic event prediction.

**Dual Heterogeneity Learning.** After event graph, context and KG merge into a blended one, it enters dual heterogeneity learning module as input. Dual heterogeneity learning is the encoder of ECO, which uses R-GCN as backbone. Depending on different type of edges, the blended graph is split into subgraphs. In each subgraph, node representations are learned according to the indegree and outdegree respectively. When aggregate node representations of different edge types, the entropy calculated will serve as the weights to distinguish various edges. And finally, the softmax layer is used as the output layer and the cross-entropy is used as the loss function to learn the score of the node.

The value of the transformed feature vector of the neighboring nodes depends on the edge type and orientation. At the same time self-loop is added as a special edge type to each node so that the node representation on layer $l + 1$ can be computed from the corresponding representation of layer $l$. The hidden representation at layer $l + 1$ can be computed as follows:

$$h_i^{(l+1)} = \sigma(\sum_{r \in R} \sum_{j \in N_i^r} \frac{1}{c_{i,\,r}} \omega_r W_r^{(l)} h_j^{(l)} + W_0^{(l)} h_i^{(l)}) \tag{8}$$

where $h_i^{(l)}$ is the hidden representation of node $i$ in layer $l$ of the model; $N$ denote the set of neighbors of node $i$ with edge type $r \in R$; $W_r$ is the weight of edge type $r$; $W_o$ is the weight of the self-connection; $\omega_r$ is the weight of $r$ learned though entropy calculation module; $c$ is a constant for normalization. $\sigma$ is an activation function to accumulate messages.

**Downstream Task Process.** Downstream task process has two branches: node classification and link prediction. For node classification, node embedding obtained get through a K-means layer to get clustering result. For link prediction, embeddings will get through DistMult(Yang et al., 2014) to obtain edge loss, and cross-entropy is used as loss function to determine whether the pair is matched.

# 4 EXPERIMENTS

After obtaining representation of nodes and edges in event garph, we model event prediction as link prediction task and node classification task. Link prediction and node classification are two typical analytical tasks on graph learning.(Xu et al., 2023) The purpose of node classification is to predict node labels in a given labeled graph, and link prediction is to predict the possibility of the existence of links between two nodes. We aim to predict fine-grained event elements, such as location and time, by link prediction and predict coarse-grained, like event types, by node classification.

## 4.1 IMPLEMENTATION DETAILS

**Dataset.** Based on the notation in 3.1, we collect real-world maritime emergency report to build **M**aritime **E**mergency **E**vents **D**ataset (**MEED**), and to conduct our experiments. The detail of our dataset is given in Table 1.

**External knowledge.** We use OpenConcepts as external knowledge to construct KG. OpenConcepts is a large-scale concept graph based on automated knowledge extraction algorithms, which has 4.4

Table 1: Statistics about **MEED**

| Number of nodes | 1.046 |
|---|---|
| Number of edges | 2.438 |
| Number of events | 381 |
| Number of event types | 9 |

Table 2: Comparison of ECO and vanilla R-GCN in link prediction

| Method | MRR | Hits@1 | Hits@3 | Hits@10 |
|---|---|---|---|---|
| R-GCN | 22.452% | 17.076% | 23.833% | 32.064% |
| ECO(our model) | 36.729% | 30.240% | 38.810% | 49.068% |

million concept entities in OpenConcepts and 13 million triples, including common entities such as places and celebrities. Concept is the essential reaction of human brain to things, which can help machines better understand natural language. Compared with traditional knowledge graph, Open-Concepts contains a large number of fine-grained concepts. For example, for the entity "Shanghai", OpenConcepts not only includes traditional concepts such as "location" and "administrative division", but also has fine-grained ones such as "areas directly under the jurisdiction of the Yangtze River Delta" and "port city".

**Training details.** We use vanilla R-GCN as benchmark. The batch size is set as 256, and the portion of graph split size is 0.5. We use Adam as the optimizer, and the learning rate is set to 1e-2. Tasks.

**Evaluation metrics.** The evaluation metrics considered for link prediction are MRR, Hits@1, Hits@3, Hits@10, and those for node classifications are Silhouette, Calinski-Harabaz index(CH), Davies-Bouldin index(DBI), purity, Adjusted Rand index(ARI), normalized mutual information(NMI), accuracy(ACC), Completeness, Homogeneity and V-Measure.

## 4.2 LINK PREDICTION

The overall results in Table 2 show that **ECO** can achieve better results than vanilla R-GCN in all metrics. It proves that the introduction of context-awareness, ontology, entropy and external knowledge can learn better events representation and boost performance of event prediction significantly.

Table 3: Ablation experiments for link prediction without entropy calculation module

| Module | | | Metric | | | |
|---|---|---|---|---|---|---|
| Context | External Knowledge | Ontology | MRR | Hits@1 | Hits@3 | Hits@10 |
| × | × | × | 22.452% | 17.076% | 23.833% | 32.064% |
| ✓ | × | × | 26.483% | 20.066% | 27.632% | 38.651% |
| ✓ | ✓ | × | 19.654% | 13.793% | 22.257% | 29.702% |
| ✓ | × | ✓ | 48.530% | 42.466% | 49.521% | 60.959% |
| ✓ | ✓ | ✓ | 37.784% | 32.060% | 39.743% | 47.824% |

Table 4: Ablation experiments for link prediction with entropy calculation module

| Module | | | Metric | | | |
|---|---|---|---|---|---|---|
| Context | External Knowledge | Ontology | MRR | Hits@1 | Hits@3 | Hits@10 |
| × | × | × | 20.867% | 15.526% | 21.271% | 31.785% |
| ✓ | × | × | 27.615% | 20.888% | 30.099% | 38.816% |
| ✓ | ✓ | × | 21.985% | 15.517% | 24.843% | 33.464% |
| × | ✓ | × | 18.194% | 11.559% | 20.430% | 30.018% |
| ✓ | × | ✓ | 45.045% | 38.559% | 47.251% | 56.687% |
| ✓ | ✓ | ✓ | 36.729% | 30.240% | 38.810% | 49.068% |

Then, we conduct a series of ablation experiments to investigate the effect of different modules used in our method. The results are shown in Table 3 and Table 4, from which we can observe:

(1) Context-awareness: On the basis of the five elementary event elements, introducing context has a significant impact on all the metrics, which prove that context-awareness is an essential part of event prediction;

(2) Entropy calculation module: Entropy can increase the hit rate of event prediction when context is also included, while it have mildly adverse influence under other circumstances;

(3) Ontology: Ontology benefits the model drastically. Compared with vanilla R-GCN method, when entropy, context and ontology are included, MRR rises by 22.593%, while Hits@n rises by 21.483% 24.623%;

(4) External knowledge: Surprisingly, external knowledge appears to diminish the advantage in all situations. Removing external knowledge improves the model performance to varying degrees.

Table 5: Results under node classification

| Metric | EG | EG+C | EG+KG | EG+C+KG | EG+C+Ont | EG+C+KG+Ont |
|---|---|---|---|---|---|---|
| Silhouette | 0.1213 | 0.1482 | 0.1697 | 0.1618 | 0.3012 | 0.2506 |
| CH | 20.5131 | 30.4529 | 29.7927 | 30.1347 | 77.7409 | 63.5362 |
| DBI | 2.3487 | 1.8582 | 1.8050 | 1.8894 | 1.2530 | 1.3532 |
| NMI | 0.1458 | 0.1787 | 0.1755 | 0.2053 | 0.2402 | 0.1732 |
| ACC | 0.0632 | 0.1500 | 0.1368 | 0.1237 | 0.1895 | 0.0658 |
| purity | 0.3579 | 0.3868 | 0.3737 | 0.4289 | 0.4211 | 0.4053 |
| ARI | 0.0593 | 0.0879 | 0.0872 | 0.0995 | 0.1141 | 0.0930 |
| Completeness | 0.1415 | 0.1739 | 0.1686 | 0.1991 | 0.2332 | 0.1666 |
| Homogeneity | 0.1504 | 0.1839 | 0.1830 | 0.2118 | 0.2476 | 0.1803 |
| V-Measure | 0.1458 | 0.1787 | 0.1755 | 0.2053 | 0.2402 | 0.1732 |

## 4.3 NODE CLASSIFICATION

Provided that entropy calculation module is included, we conduct node classification task to demonstrate our model's competence to predict the types of events. From results shown in Table 5, in which EG stands for event graph, C stands for context, Ont stands for ontology and KG stands for knowledge graph, we can draw conclusions that:

(1) Compared with predicting with event merely, the joint of context-awareness or external knowledge boosts all metrics except DBI, and the degree of improvement of event+context and event+OpenConcepts is roughly close to each other;

(2) Similarly, the combination of event+context+OpenConcepts can also boost performance. Compared with the situation that context-awareness and external knowledge are introduced separately, event+context+OpenConcepts achieves an increase in all metrics except Silhouette, CH and ACC, which indicated that this combination learn event representation effectively;

(3) The adoption of ontology gain a noticeable progress in CH. Introducing ontology to the combination of event+context+OpenConcepts is beneficial for metrics like Silhouette and CH, but it causes is slightly inferior in other metrics. Interestingly, when OpenConcepts is removed, Silhouette and CH further grow, peaking at 0.3012 and 77.7409 separately. Meanwhile, the combination of event+context+ontology also exceed other in NMI, ACC, ARI, Completeness, Homogeneity and V-Measure.

Besides, we investigate the effect of entropy calculation on event representation learning when the combination of EG+C+Onto and EG+C+KG+Onto are considered respectively. The result is demonstrated in Figure 4.

As Figure 4 depicts, the combination EG+C+Onto+Entropy achieves the best results, outperforming the remaining three cases in all metrics. Surprisingly, under the combination of EG+C+Onto, the entropy calculation module can improve the result of event representation, while under the combination of EG+C+KG+Onto, entropy calculation module slightly damages the effectiveness of representation towards different types of events.

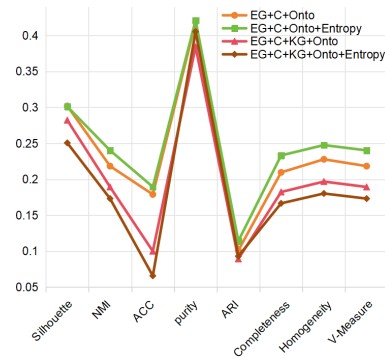

Figure 4: Effect of entropy on node classification

Moreover, we visualize the cluster result in Figure 5 to demonstrate the effect of event representation learned. It indicates that compared with only relying on event graph, introducing context, knowledge graph and ontology into representation learning can better distinguish different types of event, thus enhancing the accuracy of the prediction of event types. The visual result also shed light on a trade-

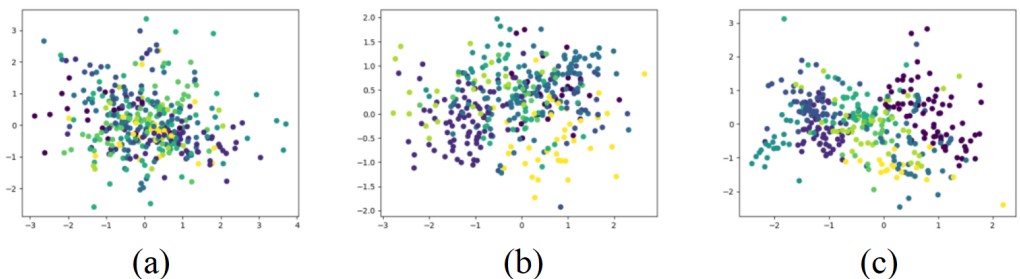

(a)            (b)            (c)

Figure 5: Visualization of the clustering results. (a) is the result conducted on event graph with vanilla R-GCN; (b) is the result of our **ECO**; (c) shows the effect of deleting external knowledge on the basis of **ECO**.
off of event representation: the combination event+context+OpenCg+ontology has larger distance between clusters, while when OpenConcepts is removed, distance within clusters shrinks.

## 4.4 DISCUSSION

Both link prediction and node classification experiments testify that our proposed model can learn better event representation for prediction, and ablation studies indicate that all the modules have different degrees of benefits on various metrics.

Introducing entropy calculation module to vanilla R-GCN helps to capture the dual heterogeneity of nodes and edges, so the representation learned can better distinguish different types of edges to make a prediction. Since considering context in event prediction is a mindset more close to human brain, it can provide more relevant details to consolidate event representation learning. Ontology have the most remarkably positive impact, because it provides abundant hierarchical knowledge between intra-event and inter-event relationships, which enables the model to learn better representation for nodes and edges of the same type.

In terms of module combination, We observed some unexpected results: the joint of external knowledge and other modules seems to diminish the advantage in link prediction. However, this does not imply that external knowledge is futile. OpenConcepts, unlike traditional external knowledge, offers the property of entities and divided them into three levels, so it render a kind of hierarchy which is similar to ontology. Thus, its advantage is less obvious in link prediction, while in the experiments of node classification, it is proved that external knowledge still has a positive effect on the learning of event embedding, which can make the representation of the same type of events more centralized.

## 5 CONCLUSION

In this paper, we propose **ECO**, a simple and effective model to learn event representation and predict events, and we establish **MEED**, a real-world maritime emergency event dataset. Our experiments indicate that **ECO** can achieve better results on various evaluation metrics. This work sheds insights that context-awareness, entropy, external knowledge and ontology are beneficial for event representation learning and can improve the performance of event prediction. There are still many promising directions remain to be explored. One aspect worth digging into is the influence of different algorithms to calculate entropy, and another aspect worth further investigating is to test whether entropy can fit in other homogeneous GNNs, to enhance their compatibility with heterogeneous graphs. Meanwhile, we intend to testify our work in more real-world scenarios. We hope our work could inspire further research interests toward the same direction.

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

## A   APPENDIX: ALGORITHM FOR ENTROPY CALCULATION MODULE

Here is the description of the algorithm we used in entropy calculation module.

This algorithm is a greedy algorithm. During the procedure, each seed neighbor will be removed to test whether its removal decreases graph entropy. Similarly, each node on the outer boundary will be added into the cluster to check whether the graph entropy decreases. The nodes on the outer boundary mean the nodes outside the cluster but having inner links to the cluster members. The process will be repeated until find a allocation with the minimal graph entropy. Hence, based on the aforesaid procedure, if the neighbors of a node can be evenly divided into inside and outside of the cluster, then this node has the highest entropy, and only the nodes that have both inner links and outer links have entropy greater than 0.

---

**Algorithm 1:** calculate node entropy and graph entropy

**Input:** $G(V, E)$
**Output:** $e\_v$

1  $e\_g \leftarrow 0$
2  $node\_state \leftarrow 0$
3  **while** $node\_state = 0$ **do**
4      Randomly select a node as seed
5      $\mathcal{A}[\,] \leftarrow$ index of node in neighbor
6      $\mathcal{B}[\,] \leftarrow$ index of node in cluster
7      $n \leftarrow sum(\mathcal{B}[\mathcal{A}])$
8      $p\_inner \leftarrow n/len(\mathcal{A})$
9      $p\_outer \leftarrow 1 - p\_inner$
10     **if** $p\_inner = 0$ *or* $p\_inner = 1$ **then**
11        $e\_v \leftarrow 0$
12     **else**
13        $e\_v \leftarrow -p\_inner * log_2 p\_inner - p\_outer * log_2 p\_outer$
14     **end**
15     $e\_g \leftarrow e\_g + e\_v$
16     $Min\_e\_g \leftarrow e\_g$
17 **end**
18 // Remove seed neighbors to minimize graph entropy.
19 **for** $neigh\_idx$ $in$ $seed\_neighbors$ **do**
20     Remove node form $\mathcal{A}$
21     Execute 4-17
22     **if** $e\_g < min\_e\_g$ **then**
23        $min\_e\_g \leftarrow e\_g$
24     **else**
25        add index to $\mathcal{A}$
26     **end**
27 **end**
28 // Add nodes from outer cluster to minimize graph entropy.
29 **for** $idx$ $in$ $neigh\_idx$ **do**
30     Add node from $\mathcal{B}$
31     Execute 4-17
32     **if** $e\_g < min\_e\_g$ **then**
33        $min\_e\_g \leftarrow e\_g$
34     **else**
35        Remove node from $\mathcal{B}$
36     **end**
37 **end**
38 **return** $e\_v$

---

