# OpenReview forum: "Entropy-enhanced context-aware event prediction based on ontology and external knowledge"
_ICLR.cc/2024/Conference — ICLR 2024 Conference Withdrawn Submission_

### Official Review · Reviewer_PDSR · 2023-10-29

**Soundness:** 1 poor
**Presentation:** 1 poor
**Contribution:** 1 poor
**Rating:** 3
**Confidence:** 4

**Summary:**

This paper presents an entropy-enhanced context-aware ontology-based method for learning representations of events in a heterogeneous graph. In an event graph with extra context information such as social environments and physical environments, the entropy of nodes and edges is utilized to characterize different types of edges. The proposed method is evaluated for downstream tasks including node classification and link prediction. This paper also constructs a maritime emergency event dataset.

**Strengths:**

1 This paper presents an interesting idea that incorporates contextual information of events (e.g., social and physical environments of an event) into a heterogeneous graph.

2 This paper presents a new dataset of maritime emergency events.

**Weaknesses:**

*** The major concern of this paper is the lack of clarity, and the quality of presentation is not good. The writing in this paper requires substantial improvement. The ideas presented are not easily discernible, mainly due to the limited context and descriptions provided. To illustrate, I will provide a few examples below.

1)	In “section 3.1.1 Event”, the description of intra-event relationship can be better presented. After introducing the quintuple, it is confusing what the event elements are  and why they are the “the foundation in a majority of event prediction tasks”. How “CAUSE” relation is formulated remains unclear.

2)	The title of the paper seems to be a combination of keywords describing the method introduced in the paper. However, motivation of using entropy is not well described. It is also not clear to me what ``ontology’’ refers to in this paper. The original words in the paper are “Event ontology is a shared, formal and explicit specification of an event class system model that exists objectively. With event ontology, semantic can be easier to share between entities, and it is also beneficial for strengthening representation and facilitating knowledge reuse.”. Maybe using an example would be helpful. It would also be nice to talk about how “devise context into event ontology” is done.

3)	In “Entropy Calculation”, it is not clear what “an induced subgraph” is and why we need it in this work. Some equations (e.g., equation 3) can be reduced or even deleted given that it is a common knowledge and it is not the main contribution of this paper. The authors claim that they “normalize the entropy of different type of edges as weights, which characterizes the dual heterogeneity of nodes and edges”. This is not clear to me what “dual heterogeneity” is and why these entropy weights can be used to characterizes the heterogeneity.

4)	In “Dual Heterogeneity Learning”, it is unclear to me how “context and kG” are merged into one graph. It is also not clear how embeddings of nodes in this graph are computed for equation 8.

Regarding the methodology, this paper misses important details describing the technical contributions in the proposed framework.


*** From the perspective of evaluation, it is surprising to see no other state-of-the-art models are compared with in node classification and link prediction (only R-GCN is included). It is mentioned that graph neural networks have been developed for event prediction. I believe many existing models can be applied to this task, but no explanation is provided for the lack of comparison with other baseline approaches.

*** The authors claim that one of the contributions is the establishment of a new real-world dataset. However, very limited details are provided for this dataset. What is the major purpose of this dataset? What are the nodes/edges in this data?

*** Overall, I think the presentation and writing of this paper could be further improved. It is hard to judge the novelty of this work given limited clarity. More in-depth analysis on unique challenges in the dataset and results could be better delivered. I think the content should focus on explaining the major contributions in this paper instead of existing technologies (e.g., entropy or graph neural networks).

**Questions:**

The questions can be found in the weaknesses.

---

### Official Review · Reviewer_Mc25 · 2023-10-31

**Soundness:** 2 fair
**Presentation:** 1 poor
**Contribution:** 2 fair
**Rating:** 3
**Confidence:** 4

**Summary:**

This paper introduces a novel method for event prediction in Natural Language Processing (NLP), named Entropy-enhanced Context-aware Ontology-based model (ECO). The authors focus on addressing the issues present in homogeneous graph-centric approaches and propose to leverage entropy, context-awareness, event ontology and external knowledge to improve the efficiency and accuracy of event prediction.
However, I have several concerns about its novelty, the lack of comparison with other works, poor language quality, and missing theory. These drawbacks call for fundamental improvements, and I suggest that the authors revise the paper focusing on the aforementioned points before resubmission.

**Strengths:**

- The studied problem for handling the dual heterogeneity in event graphs is interesting.
- The creation of an ontology-based model integrating realistic contexts into event graphs seems a useful addition to the field.
- Introduction of a new maritime dataset, MEED, for event prediction is a positive contribution, providing a potential new benchmark for future works.

**Weaknesses:**

- The paper's approach appears to be a simple combination of several established techniques in the field. A fusion of entropy-enhanced learning, ontology-based modeling, context-awareness, and use of external resources, which subtracts from the novelty of this contribution.
- It seems that the authors have not promised they will release the codes and the new dataset. This lack of transparency makes it difficult for other researchers to replicate and validate the presented model and dataset, which may prevent other researchers build upon or learn from this work effectively.
- The experimental results fail to offer a comprehensive picture as there is no comparison with other state-of-the-art models. By only showing how ECO outperforms its variants in ablation style, the paper lacks convincing arguments on how it could be superior to other existing methods.
- Writing quality of the paper is poor. Numerous typographical and grammatical errors affect readability and understanding. For example, in the Introduction Section, "NLP.(Du et al., 2022)" --> "NLP (Du et al., 2022)."; "an car" --> "a car".
- The paper heavily relies on the concept of entropy, but theoretical insight into the entropy calculation module and how it addresses heterogeneity is scant.

**Questions:**

Please see weakness above.

---

### Official Review · Reviewer_gtQg · 2023-11-04

**Soundness:** 2 fair
**Presentation:** 2 fair
**Contribution:** 2 fair
**Rating:** 3
**Confidence:** 5

**Summary:**

This paper focuses on a very interesting problem of event prediction. It highlights several challenges in this problem and designs an Entropy-enhanced Context-aware Ontology-based model (ECO). It considers the heterogeneity of the event graph, the context, and the external knowledge. It also builds a new dataset of MEED in the domain of maritime emergency events.

**Strengths:**

1.	The research question of event prediction, especially in the vertical domain of maritime emergency management, is interesting and valuable.
2.	The proposed event dataset of MEED brings a new application scenario to the event research community. This resource is meaningful and helpful.
3.	The key aspects that the paper mentions, i.e., the context and external knowledge are reasonable in the problem of event forecasting.

**Weaknesses:**

1.	This paper has missed some key reference papers from multiple aspects, including 1) works on temporal knowledge graph, such as RE-NET[1] and RE-GCN[2], which are all based on heterogeneous graphs (where nodes and edges are of various types) for event forecasting; 2) works that use contexts for event forecasting, such as CMF[3] and SeCoGD[4]. Even though the dataset of this paper is newly introduced and a little bit different from those of the previous works as listed above [1-4], the difference is not that essential. Therefore, the main arguments of this paper are questionable: 1) current works are only homogeneous; and 2) current works have not considered contexts.
2.	The motivation for using this entropy-enhanced method is not well presented, especially in the introduction. It is not that obvious or intuitive to use this novel technique, thus the motivation is necessary to be presented in the introduction.
3.	In line with the w1, the baseline methods are insufficient. It should compare with the works including but not limited to [1-4]. In terms of non-temporal KG methods, CompGCN [5] is also a potential well-performing baseline. In terms of the downstream task, in addition to DistMult, more effective decoders, such as ConvE[6] or ConvTransE[7], could be utilized and implemented.
4.	In terms of problem formulation, the main objective of this paper is event prediction, which is implemented as two tasks of node classification and link prediction. However, from my point of view, it is better to zoom in on certain forecasting-centric settings, for example, there are follow and cause relations between the events as shown in the definition, it is more interesting to predict the next event or the caused event given a part of the follow and cause triplets.

* [1] Recurrent Event Network: Autoregressive Structure Inference over Temporal Knowledge Graphs
* [2] Temporal Knowledge Graph Reasoning Based on Evolutional Representation Learning
* [3] Understanding Event Predictions via Contextualized Multilevel Feature Learning
* [4] Context-aware Event Forecasting via Graph Disentanglement
* [5] Composition-based Multi-Relational Graph Convolutional Networks
* [6] Convolutional 2D Knowledge Graph Embeddings.
* [7] End-to-End Structure-Aware Convolutional Networks for Knowledge Base Completion.

**Questions:**

See the weaknesses.

---

### Official Review · Reviewer_JEH5 · 2023-11-05

**Soundness:** 2 fair
**Presentation:** 3 good
**Contribution:** 3 good
**Rating:** 5
**Confidence:** 4

**Summary:**

The authors constructs a Maritime Emergency Events Dataset (MEED) for event prediction. They propose a context-aware (e.g. weather etc.) and external knowledge (OpenConcepts KG for entity information) integrated ontology representation in constructing the dataset from  maritime reports. In addition, they propose a graph node and edge entropy (with a heuristic algorithm to minimize entropy by removing and adding edges) based R-GCN learning method for event representations.

They conduct ablations of representation and learning in node classification/link prediction ("We aim to predict fine-grained event elements, such as location and time by link prediction; and predict coarse-grained, like event types, by node classification).

**Strengths:**

1. Dataset - The authors constructs a Maritime Emergency Events Dataset (MEED) for event prediction.

2. Event ontology extensions - They propose a context-aware (e.g. weather etc.) and external knowledge (OpenConcepts KG for entity information) integrated ontology representation in constructing the dataset from maritime reports.

**Weaknesses:**

1. Unclear effectiveness and complexity - In addition, they propose a graph node and edge entropy (with a heuristic algorithm to minimize entropy by removing and adding edges) based R-GCN learning method for event representations.
- The experiments show without entropy calculation module results are better (48 vs 45 MRR in Table 3 and 4).
- The algorithm in the appendix is not a clear enough pseudo-code for "minimizing entropy by greedily removing or adding nodes" (the neighbors and seeds are not clearly defined, what happens to the subgraph connected to those nodes or are they only leaf nodes?)
- R-GCN would automatically learn weights and feature vectors, thereby the whole idea of feature engineering entropy minimization to down-weight certain nodes or edges seem worse than leveraging the neural R-GCN learning to figure it out.

2. Unclear details of experiment tasks - They conduct ablations of representation and learning in node classification/link prediction ("We aim to predict fine-grained event elements, such as location and time by link prediction; and predict coarse-grained, like event types, by node classification).
- It is unclear exactly what is the detailed experimental formulation is unclear for both learning and evaluation. In this dataset, what are the types for node and link prediction (how many classes/labels and what are they), are they predicted jointly or separately. The paper only describes this in one high-level sentence quoted above.
- Are event graphs being combined or treated separately? (one for each event in the dataset). Why are there only 1 nodes and 2 edges (in the Table 1 stats).
- What about the "Follow/Cause" etc between events? Is that represented in this dataset? And does event prediction try to predict those?

3. Overall results and other baselines (beside vanilla RGCN) - The overall result is ~30-48 on link prediction, and very low accuracy on node classification (7%). Existing ontology works should have been considered baselines on this task and dataset, rather than vanilla R-GCN alone. It is also not clear whether the RGCN performance is less because of which ablation component.

4. (Minor) External knowledge and way of integrating in the ontology seems ineffective - Why isn't it better to attach the external triples as links to the event participants entities directly, rather than separately onto the context directly from the event/context elements node? (Fig 2). Is it that there are edges between say parameters_of_vessel (in context) and vessel object (in event objects)?

**Questions:**

See weakness - the main issues seem in the learning part, experimental methodology or task details. I appreciate MEED dataset and ontological considerations.